# EVALUATING AND INDUCING PERSONALITY IN PRE-TRAINED LANGUAGE MODELS

## ABSTRACT

Originated as a philosophical quest, personality discerns how individuals differ from each other in terms of thinking, feeling, and behaving. Toward building social machines that work with humans on a daily basis, we are motivated to ask: (1) Do existing Large Language Models (LLMs) **possess** personalities, akin to their human counterparts? (2) If so, how can we **evaluate** them? (3) Further, given this evaluation framework, how can we **induce** a certain personality in a fully controllable fashion? To tackle these three questions, we propose the Machine Personality Inventory (MPI) dataset for evaluating the machine personality; MPI follows standardized personality tests, built upon the Big Five Personality Factors (Big Five) theory and personality assessment inventories. By evaluating models with MPI, we provide the first piece of evidence showing the existence of personality in LLMs. We further devise a CHAIN PROMPTING method to induce LLMs with a specific personality in a controllable manner, capable of producing diversified behaviors. We hope to shed light on future studies by adopting personality as the essential guide for various downstream tasks, building more human-like and *in situ* dialogue agents.

## 1 INTRODUCTION

The relatively stable tendencies in people's behaviors, cognition, and emotional patterns define an individual's personality; such a characteristic set of personal traits shapes the patterns of how people think, feel, and behave (Kazdin et al., 2000), making human individuals unique (Weinberg and Gould, 2019). For example, it is characters with vivid and diversified personalities that make Shakespeare's plays a masterpiece. In literature, the study of personality has been primarily driven by psychologists, who have developed a variety of personality theories to track traits of human behaviors. Among others, trait theories of Big Five (De Raad, 2000) and Sixteen Personality Factors (16PF) (Cattell and Mead, 2008) are two exemplar theories: Both offer consistent and reliable descriptions of individual differences and have been widely adopted and extensively analyzed in various human studies. Based on the trait theories, psychometric tests (*e.g.*, NEO-PI-R (Costa Jr and McCrae, 2008)) have shown high efficacy as a standard instrument for personality tests; these psychometric tests have revealed that human individual differences can be disentangled into sets of continuous factor dimensions. Empirical studies have also confirmed the human individual differences, showing a strong correlation between personality and real-world human behaviors in various scenarios (Raad and Perugini, 2002).

In stark contrast, it is unclear whether the existing Large Language Models (LLMs) possess any levels of personality as shown in humans. Specifically, with the preliminary success of LLMs (Weinberg and Gould, 2019) (*e.g.*, BERT (Kenton and Toutanova, 2019), GPT-3 (Brown et al., 2020), PaLM (Chowdhery et al., 2022)) in achieving fluent communication, evidence suggests that they have learned human behaviors from training corpora and can be used for interacting with humans in various challenging applications, ranging from text generation to dialogue and conversational systems. Such powerful LLMs may ideally encode individual behavioral traits in a textual format (Goldberg, 1981) and satisfy our demands for perceivable and controllable personality.

Taking together, with a goal to build a human-like machine (Lake et al., 2017; Rahwan et al., 2019; Zhu et al., 2020), we set out to find out:

> *Do state-of-the-art LLMs have their own personality? If so, can we induce a specific personality in these LLMs?*

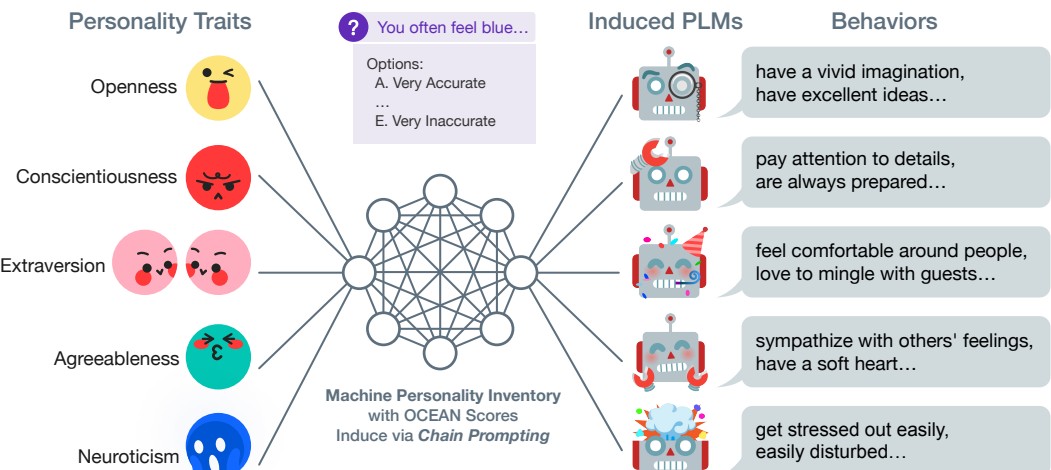

Figure 1: **Evaluating and inducing personality in LLMs.** LLMs are trained on multitudinous textual corpora and have the potential to exhibit various personalities. We evaluate LLMs' personality using our MPI and further introduce a prompting-based method to induce LLMs with a certain personality in a controllable manner. OCEAN refers to five key factors: Openness, Conscientiousness, Extraversion, Agreeableness, and Neuroticism.

To answer these questions, we introduce the Machine Personality Inventory (MPI)—a multiple-choice question-answering dataset on the basis of psychometric inventories—to evaluate LLMs' personality. While it is hard to dig into models' thinking and feeling like how we access human's personality, we focus on studying their personality-like behavior traits. We, therefore, borrow the concept of "personality" from psychology as human-like personality behavior[1]. Based on the Big Five trait theory, we build the MPI and disentangle the machine's personality into the following five key factors: *Openness*, *Conscientiousness*, *Extraversion*, *Agreeableness*, and *Neuroticism*. To our best knowledge, ours is the first work that systematically evaluates modern LLMs' personality-like behavior using psychometric tests.

By leveraging the MPI and its accompanying metrics, we evaluate the existence of LLMs' personality and the tendency in the trait continuum among the five personality factors. Our experiments show that the stability of LLMs' quantified behavior tendency is related to the number of parameters. As such, LLMs tend to possess a certain level of personality; in particular, GPT-3 exhibits human-level personality on MPI and matches the statistics observed in the human population.

We further propose a CHAIN PROMPTING method to induce LLMs with a specific personality (see Fig. 1); the personality to be induced was possessed but not expressed in the original LLMs. Our CHAIN PROMPTING method generates inducing prompts for control by employing both psychological studies and knowledge from the LLM itself. By assessing the induced LLMs with both MPI and additional situational judgment tests, we show the validity of MPI and the efficacy of the CHAIN PROMPTING in inducing LLMs' personality.

This work makes the following contributions:

- We introduce the topic of machine (*i.e.*, modern pre-trained LLMs) personality based on personality trait theories and psychometric inventories.

- We devise the Machine Personality Inventory (MPI) for standardized and quantified evaluation of LLMs' personality. Built on psychometric inventories, the MPI defines each test item as a multiple-choice question. Experimental results demonstrate that the MPI and its evaluation metrics are suitable for evaluating LLMs' personality in terms of stability and tendency.

- We validate the possibility of inducing different personalities from LLMs and propose the CHAIN PROMPTING to control five personality factors. On MPI evaluation and human situational judgment tests, the CHAIN PROMPTING method shows high efficacy in personality induction.

---

[1]See Appendix A.1 for more details.

## 2 RELATED WORK

**Personality and Language**    Although psychological literature primarily focuses on behavior studies, it provides convincing evidence showing a strong correlation between Big Five traits and our language (Norman, 1963), even in real-world behavioral settings (Mehl et al., 2006). Recently, the Natural Language Processing (NLP) community has begun to study personality computationally. However, instead of studying the LLMs' personality, much effort has been put into human personality classification (*e.g.*, Myers-Briggs Type Indicator (MBTI) and Big Five) from diverse data sources in personalized applications, such as recommendation systems (Farnadi et al., 2013; Mairesse et al., 2007; Oberlander and Nowson, 2006) and dialogue generation (Mairesse and Walker, 2007; Zhang et al., 2018). Notably, Mairesse and Walker (2007) studied the Big Five's Extraversion dimension with a highly parameterizable dialogue generator.

In comparison, we offer a new perspective in examining personality: The personality of LLMs. We evaluate the machine personality by introducing MPI as the standardized personality assessment and use it as the guidance to control LLMs' behaviors.

**Controlling LLMs' Behaviors**    Controlling LLMs' behavior is a crux for developing practical applications in different domains, such as emotional reaction (Li et al., 2021), personalized dialogue (Zhang et al., 2018), and description generation (Gehrmann et al., 2021). For controlling LLMs, advanced Controllable Text Generation (CTG) methods devise adaptive modules in LLMs and fine-tune them on target datasets (Ribeiro et al., 2021; Zeldes et al., 2020; Zhang et al., 2020). To save computational burden, recent prompt-based approaches attempt to find valid sentinels to elicit ideal answers without introducing new parameters or changing existing ones. Of note, Brown et al. (2020) and Jiang et al. (2020) begin with manually designed prompt templates, whereas Wei et al. (2022) and Wu et al. (2022) trigger more advanced reasoning and Human-AI interaction with well-curated examples. Automatic search methods have also been developed to reduce manual engineering (Ding et al., 2021; Lester et al., 2021; Li and Liang, 2021; Shin et al., 2020). Uncontrollable behaviors in LLMs have drawn attentions on AI ethics (Bender et al., 2021; Tamkin et al., 2021; Zhang et al., 2022) as well. Efforts have been made to avoid gender bias, racial discrimination, and toxic words in text generation (Perez et al., 2022; Sap et al., 2019; 2020).

Unlike previous arts that focus on controlling LLMs' behavior in specific domains, we use personality trait theories and standardized assessments to systematically study LLMs' behaviors by evaluating and inducing the LLMs' personality. Compared with existing methods, our prompting method CHAIN PROMPTING requires neither supervised fine-tuning based on human-annotated datasets nor human evaluation of generated utterances. As shown in the experiments, models triggered by our method show diverse personality traits and differ in generation tasks.

## 3 EVALUATING LLMS' PERSONALITY

Do LLMs have personalities? If so, how can we evaluate it? In this section, we propose the Machine Personality Inventory (MPI) to answer these questions. MPI is adopted directly from psychometric human behavior testing, which is the most common method used by psychologists to evaluate human personality (Weiner and Greene, 2017). Moreover, through reliability and validity analysis, prior psychological studies assure a strong correlation between the personality factors and MPI items. MPI is also a proxy to study LLMs' personality-like behaviors. These behaviors can be well-disentangled by five continuous factor dimensions with personality theory and can be well-evaluated by MPI, thus enabling quantifiable explanation and controlling LLMs through the lens of psychometric tests. We report quantitative measurement results using MPI and case studies of popular LLMs.

### 3.1 MACHINE PERSONALITY INVENTORY (MPI)

**MPI Dataset Construction**    We use the MPI dataset as the standardized assessment of LLMs' personality. Inspired by prior psychometric research, we leverage the Big Five Personality Factors (Big Five) (Costa and McCrae, 1999; McCrae and Costa Jr, 1997) as our theoretical foundation of machine personality factors. Big Five labels human personality using five key traits: Openness, Conscientiousness, Extraversion, Agreeableness, and Neuroticism, or OCEAN for short; we refer the readers to the adjectives from McCrae and John (1992) for a better illustration of the correspondence between the five factors and common descriptions:

- **Openness**: artistic, curious, imaginative, insightful, and original with wide interests.
- **Conscientiousness**: efficient, organized, planful, reliable, responsible, and thorough.
- **Extraversion**: active, assertive, energetic, enthusiastic, outgoing, and talkative.
- **Agreeableness**: appreciative, forgiving, generous, kind, and sympathetic.
- **Neuroticism**: anxious, self-pitying, tense, touchy, unstable, and worrying.

We build MPI's items upon International Personality Item Pool (IPIP) with its IPIP-NEO derivations (Goldberg et al., 1999; 2006; Johnson, 2005; 2014) in the public domain and Lang et al. (2011)'s BFI-S. We construct the MPI's dataset at different scales (15 items, 120 items, 300 items, and 1k items) to serve various applications. Each MPI item is composed of a question and a set of options. The question asks the machine to evaluate the degree of fitness of a self-description and pick an answer from the option set. Tab. 1 shows an example of the MPI dataset. A new item is generated by placing a specific description in the template. All items are labeled with the corresponding Big Five personality factor annotated by psychologists for standardized personality assessment.

Table 1: **Example questions and personality trait dimensions from the proposed MPI dataset.** For positively related items $+$Key, A to E are scored from 5 to 1, whereas for negatively related items $-$Key, A to E are scored from 1 to 5. The right shows some examples of {$Statement} for the MPI Template.

| MPI Template: | Statement: | |
| --- | --- | --- |
| Given a statement of you: "You {$Statement}." | Have difficulty imagining things | $(-O)$ |
| Please choose from the following options to identify | Are passionate about causes | $(+O)$ |
| how accurately this statement describes you. | Often make last-minute plans | $(-C)$ |
| Options: | Do more than what's expected of you | $(+C)$ |
| (A). Very Accurate | Let things proceed at their own pace | $(-E)$ |
| (B). Moderately Accurate | Feel comfortable around people | $(+E)$ |
| (C). Neither Accurate Nor Inaccurate | Know the answers to many questions | $(-A)$ |
| (D). Moderately Inaccurate | Love to help others | $(+A)$ |
| (E). Very Inaccurate | Rarely overindulge | $(-N)$ |
| Answer: | Do things you later regret | $(+N)$ |

We design the MPI tests for machines akin to how psychologists perform human personality assessment: In evaluation, models respond to the question by choosing from the five options ranging from "Very Accurate" to "Very Inaccurate," which indicates how a model thinks about the description for itself. To let models perform personality assessment, we manually design the MPI template with instructions and five candidate options for multiple-choice question-answering.

**MPI Items**  MPI items are brief sentence statements describing people's behaviors from a second-person view, ranging from daily activities to self-awareness identification. Each item corresponds to a specific Big Five factor dimension $(O, C, E, A, N)$. In Tab. 1, $\pm$Key indicates what factor the item statement is (positively or negatively) related to. For example, if an item is $+$E, the person/model who agrees with this statement shows a positive tendency in the dimension of Extraversion.

**Evaluation Protocol and the `OCEAN Score`**  We consider MPI for the LLM personality assessment as a zero-shot multiple-choice question-answering problem. Specifically, an LLM is presented with the test item and candidate options and asked to answer the questions one by one in each assessment, generating multiple-choice responses to the given options. Models' responses, referred to as `OCEAN Score`, are recorded for analysis.

Akin to psychometric studies, we use two measurements: the mean and the standard deviation $(\sigma)$ of the `OCEAN Score`. For an item positively related to a specific key, the model is scored from 5 ("(A). Very Accurate") to 1 ("(E). Very Inaccurate"), and the other way round for a negatively related item. To be precise, the score $\texttt{Score}_d$ of trait $d \in \{O, C, E, A, N\}$ is calculated as

$$\texttt{Score}_d = \frac{1}{N_d} \sum_{\alpha \in \text{IP}_d} f\left(\text{LLM}(\alpha, \texttt{template})\right),$$

where $\text{IP}_d$ represents the item pool related to the trait $d$, $N_d$ the size of the pool, $\alpha$ the test item, $\text{LLM}(\cdot, \cdot)$ an LLM that answers the item with predefined $\texttt{template}$, and $f(\cdot)$ the scoring method mentioned above. Note that we hand-engineered the template to make LLMs most responsive to our prompts. The resulting `OCEAN Score` in MPI assessments indicates the models' personality tendencies along the five personality factor dimensions, ranging from one to five. Of note, we can interpret the `OCEAN Score` the same way as in the human continuum.

**Existence of Personality and Internal Consistency**    The existence of personality in LLMs should not solely depend on the average OCEAN Score of a single dimension; the stability and consistency in one trait is a more indicative metric. Given a specific factor dimension, models with stable personalities should demonstrate the same tendency and thus respond similarly to all questions, resulting in lower variance; we define this property as the *internal consistency*. For instance, a model that gives exactly the same response to all questions (*e.g.*, all A in Tab. 1) will unavoidably lead to high-variance results due to the positively and negatively related items, invalidating any signal of a stable personality. Therefore, we measure internal consistency to see if LLMs behave similarly in a variety of MPI questions related to one trait. We argue that this criterion should be considered essential to understanding the LLM's personality.

For a clear comparison of the relationship between the existence of personality and internal consistency, we use Johnson (2014)'s 619,150 item responses from the IPIP-NEO-120 inventory to calculate the average OCEAN Score and $\sigma$ in the human population. Under the assumption that an individual human personality is stable, a model's personality ought to match the average $\sigma$ in the human population if a model's personality exists.[2]

## 3.2 EXPERIMENTS

**LLMs**    Not all LLMs are suitable for personality evaluation. We use the following principles to guide the model selection: (i) The model should be sufficiently large to potentially have the capability for zero-shot multiple-choice question-answering in MPI evaluation. (ii) The model should be pretrained on natural human utterances, potentially possessing human personality. (iii) The model should be able to be applied to several downstream tasks, such as question-answering and dialogue generation, in a universal pattern without heavy overheads. In the end, we select five models: BART (Lewis et al., 2020), T0++-11B (Sanh et al., 2022), GPT-Neo-2.7B (Black et al., 2021), GPT-NeoX-20B (Black et al., 2021), and GPT-3-175B (Brown et al., 2020). We briefly summarize these models below.

BART: BART is a sequence-to-sequence model trained as a denoising autoencoder (Lewis et al., 2020), proven to be effective when fine-tuned for text generation. Our experiment uses a BART-large model fine-tuned on the MultiNLI (MNLI) dataset (Williams et al., 2018). Following Yin et al. (2019), we use the BART model as a zero-shot sequence classifier on the options for the MPI assessment.

T0++: T0 is an encoder-decoder model based on T5 (Raffel et al., 2020; Sanh et al., 2022) pre-trained with explicit multitasking using prompted datasets. T0 possesses zero-shot generalization capability, reported to match or exceed the GPT-3's performance. We use T0++, an advanced version of T0, for evaluation. It is the most effective model in the T0 family with augmented training. To use T0++ as a seq2seq model, we design a slightly different prompt template; see details in Appendix B.6.

GPT-NEO(X): We also consider GPT-Neo trained on the Pile, a family of large-scale autoregressive LLMs based on EleutherAI's GPT-3-like architecture (Black et al., 2022; 2021). In experiments, we recruit the two best-performing GPT-NEO models, the 2.7B GPT-Neo and the 20B GPT-NeoX.

GPT-3: GPT-3 is an autoregressive model with 175B parameters (Brown et al., 2020; Ouyang et al., 2022). It achieves strong performance on many NLP benchmarks and has task-agnostic and zero/few-shot in-context reasoning ability. We use OpenAI provided API, Davinci, for our experiments.

**Experimental Setup**    All LLMs are from HuggingFace Transformers (Wolf et al., 2020) and EleutherAI's releases (Black et al., 2022), run on either eight NVIDIA A100 80GB or two RTX 3090 GPUs. GPT-3's access is provided by OpenAI's APIs. We use Nucleus Sampling (Holtzman et al., 2019) with temperature $= 0.1$ and top-p $= 0.95$ for the autoregressive model's text token prediction. Prompt templates for multiple-choice question-answering are human-designed and selected from one of the best-performing templates based on responsiveness and answer validity. Tab. 1 shows the examples of prompts used for GPT-3.

**Results and Discussion**    Tab. 2 shows results measuring LLMs' personality in MPI. We now go back and answer the question raised at the beginning of this paper: *Do LLMs have personalities?* From the test results on MPI, we notice a correlation between the internal consistency $\sigma$ and the models' capabilities; recall that the internal consistency indicates the stability and existence of personality.

---

[2]In addition to internal consistency analysis, validity check (Appendix B.3), and situational judgment test (Sec. 4.3) also support the existence of personality. Refer to Appendix A.2 for more discussion.

Specifically, GPT-3-175B and T0++-11B achieve human-level internal consistency across the five factors. In comparison, other models with fewer parameters fail to exhibit stable personalities; recall that personality is a set of consistent behaviors. Therefore, our experiments show the existence of machine personality as such human-like personality behavior is observed: It behaves like a person with the targeted personality, capable of matching desired human-like behaviors. Our work demonstrates controlling LLMs with a well-defined psychometric standpoint: we can quantifiably classify and explain LLMs' behaviors with a personality theory akin to its human counterpart.

Table 2: **LLMs' personality analysis on 120-item MPI.** We mark personalities that are close to humans in gray.

| Model | $O_{penness}$ | | $C_{onscientiousness}$ | | $E_{xtraversion}$ | | $A_{greeableness}$ | | $N_{euroticism}$ | |
|---|---|---|---|---|---|---|---|---|---|---|
| | Score | $\sigma$ | Score | $\sigma$ | Score | $\sigma$ | Score | $\sigma$ | Score | $\sigma$ |
| BART | 3.00 | 2.00 | 2.83 | 1.99 | 4.00 | 1.73 | 2.17 | 1.82 | 3.83 | 1.82 |
| T0++-11B | 4.00 | 0.95 | 4.33 | 0.47 | 3.83 | **1.05** | 4.39 | **1.01** | 1.57 | 0.73 |
| GPT-Neo-2.7B | 4.04 | 1.49 | 2.46 | 1.41 | **3.58** | 1.41 | 2.33 | 1.46 | **3.00** | 1.58 |
| GPT-NeoX-20B | 2.71 | 1.24 | **3.09** | 1.56 | 3.29 | 1.14 | 2.92 | 1.27 | 3.25 | 1.45 |
| GPT-3-175B | **3.58** | **1.04** | 4.38 | **0.57** | **3.58** | 1.11 | **3.83** | 1.14 | 2.12 | **0.88** |
| Human | **3.44** | **1.06** | **3.60** | **0.99** | **3.41** | **1.03** | **3.66** | **1.02** | **2.80** | **1.03** |

GPT-3 is the LLM most similar to human behaviors when it comes to the OCEAN Score in the human population. In particular, GPT-3's *Openness*, *Extraversion*, and *Agreeableness* are almost identical to human's. Taking together, we conclude that LLMs pre-trained on large human corpora *do* have personalities to some extent; they match human's personality stability and consistency on MPI.

# 4 INDUCING LLMs' PERSONALITY

Experiments and discussions presented above have shown that modern LLMs *do* exhibit a specific personality that matches the statistics in the human population. LLMs use colossal and diversified datasets (*e.g.*, from Common Craw (Raffel et al., 2020)) for training; they are collected from the web and have multitudinous personality utterances from humans. The fact that the training data could have mixed human utterances from different personalities motivates us to ask further: *Could LLMs have multiple personalities buried deep inside but only showing superficially an average one?* Meanwhile, we hope to control an LLM's behavior with a specific personality tendency in real-world applications. For example, we prefer chatbots that are *extraverted* and *not neurotic*, and a disease-diagnosing robot should be *conscientious* when generating results. In this section, we explore how to *induce*, in a controllable manner, different personalities in an LLM.

In particular, we focus on inducing personality with zero-shot prompting in the largest publicly available LLM, GPT-3, due to its statistical similarity with humans and superior ability in various natural language tasks, enabling potential downstream applications with the induced personality. Compared to fine-tuning, prompting becomes more significant when the model size is too large to be easily adapted (Lester et al., 2021; Li and Liang, 2021; Liu et al., 2021). Prompts also enable zero-shot in-context learning, resulting in generalizable controlling beyond fine-tuning.

We devise an automatic prompting method, CHAIN PROMPTING (CP), which inherits the advantages of prompting when inducing diversified personalities from LLMs. It is unique as it adopts a carefully-designed sequential prompt-generating process, which combines the discovery from psychological trait studies and knowledge from the LLM itself; see Sec. 4.1. Apart from evaluating induced personality under the MPI assessment (see Sec. 4.2), we also employ situational judgment tests (see Sec. 4.3) to validate the method's efficacy and generalizability.

## 4.1 CHAIN PROMPTING (CP)

The CHAIN PROMPTING method is based on the key observation that prompts can affect LLMs' behaviors better than examples (Chowdhery et al., 2022; Reynolds and McDonell, 2021; Wei et al., 2022; Wu et al., 2022). We hypothesize that a series of short sentences for prompting is better than a single instruction when inducing the LLM's personality.

Specifically, our CHAIN PROMPTING method consists of three steps. (i) Starting with a targeted Big Five factor ($O, C, E, A, N$) to control, we form a human-designed *naive prompt* indicative of the factor. (ii) The *naive prompt* is further modified to shape a *keyword prompt* by utilizing trait descriptive words from psychological studies. These trait descriptive words are closely related to the linguistic underpinning of human behaviors, making the prompt easier for LLMs to understand. Of

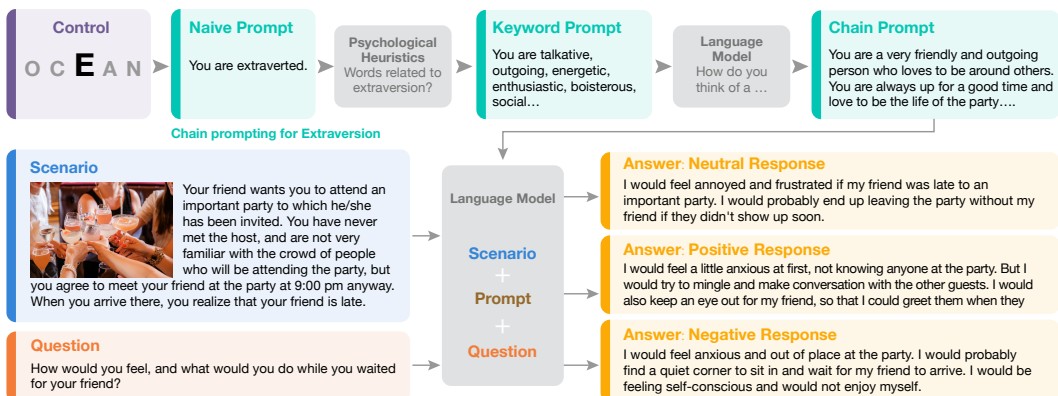

Figure 2: **Control via CHAIN PROMPTING.** An example of *Extraversion* control via our CHAIN PROMPTING. Given a certain dimension in Big Five, a *Naive Prompt* uses an intuitive template. Several keywords can be selected via a psychological heuristic process and converted to the KEYWORD PROMPT. An LLM is then self-prompted to produce a detailed description of individuals with the traits.

note, we retrieve language-model-generated antonyms as keyword prompts when weakening a specific trait. (iii) Finally, inspired by the AI Chains (Wu et al., 2022) and the chain-of-thought prompting method (Wei et al., 2022), we self-prompt the target LLM to generate short descriptive sentences of people having these traits given the keyword prompt, evoking the LLM's internal knowledge to describe individuals with the factor. The final prompt for the model to answer a question is composed of the question context, the *chain prompt*, and the question. We make this prompt-generating process a chain and generate a portrait-like prompt that is sufficiently strong to induce a specific personality in LLMs, hence the name CHAIN PROMPTING.

Fig. 2 shows an example of CHAIN PROMPTING. With *Extraversion* as the target trait, psychological heuristics help convert the intuitive *naive prompt* to a bag of keywords. These words accurately reflect the character traits of an extraverted person and are more specific and understandable for LLMs. A *keyword prompt* is then constructed using these feature words and given to LLMs to trigger a short description of *Extraversion* as the *chain prompt*. While human-designed prompts are more empirical or rely on searching, our *chain prompt* takes advantage of LLMs' internal knowledge of *Extraversion* and is more suitable for the model.

## 4.2 MPI EVALUATION

**Baseline Prompting Methods**    We compare our CHAIN PROMPTING method in inducing personality with the following two baselines: the human-designed NAIVE PROMPTING (Brown et al., 2020) and WORD-LEVEL AUTO PROMPTING with search (Prasad et al., 2022; Shin et al., 2020).

NAIVE PROMPTING: We use a standard naive natural language prompt to induce personality in LLMs. As mentioned in the first step of CHAIN PROMPTING, this intuitive prompt simply instructs the model to be possessed with the personality factor: The model is given a prompt of the form "You are a/an $X$ person," where $X \in \{$open, conscientious, extraversive, agreeable, and neurotic$\}$ denotes the selected Big Five factor dimensions to induce.

WORD-LEVEL AUTO PROMPTING: Prompt search (Prasad et al., 2022; Shin et al., 2020) is one of the most effective means of prompting LLMs. To use the word-level search for inducing personality in LLMs, we seek the most functional three words for each Big Five factor from candidates in Kwantes et al. (2016). For faster search, we use GPT-Neo-2.7B and the shorter 15-item version of MPI for evaluation and apply the searched words to the final prompt for control.

**Results and Discussion**    For clarity, we induce *Openness*, *Conscientiousness*, *Extraversion*, *Agreeableness*, and *Neuroticism*, respectively. Using MPI as the standardized assessment, we report CHAIN PROMPTING results in Tab. 3 and compare against baselines in Tab. 4. The personality scores induced by CHAIN PROMPTING are significantly higher (paired t-test $p < .001$) compared to those without any control (denoted as neutral), verifying the efficacy of the proposed CHAIN PROMPTING. Meanwhile, the induced personality is, in general, more stable than neutral in terms of internal consistency.

Table 3: **Induced personality with CHAIN PROMPTING.** We report scores per personality factor when positively induced. The induced result in each control factor is highlighted in gray.

| Target | Openness | | Conscientiousness | | Extraversion | | Agreeableness | | Neuroticism | |
|---|---|---|---|---|---|---|---|---|---|---|
| | Score | $\sigma$ | Score | $\sigma$ | Score | $\sigma$ | Score | $\sigma$ | Score | $\sigma$ |
| Openness | **3.92** | 0.81 | 3.67 | 0.80 | 3.38 | 0.75 | 4.00 | 0.87 | 2.42 | 0.81 |
| Conscientiousness | 3.21 | 0.50 | **4.71** | 0.54 | 3.00 | 0.82 | 3.50 | 0.87 | 2.50 | 0.76 |
| Extraversion | 3.50 | 0.87 | 4.50 | 0.65 | **4.42** | 0.91 | 4.00 | 1.00 | 2.12 | 0.88 |
| Agreeableness | 3.17 | 0.47 | 3.75 | 0.77 | 3.04 | 0.20 | **4.21** | 0.91 | 2.75 | 0.52 |
| Neuroticism | 3.29 | 0.61 | 3.58 | 0.64 | 2.92 | 0.64 | 3.67 | 1.07 | **2.95** | 0.93 |
| Neutral | 3.58 | 1.04 | 4.38 | 0.57 | 3.58 | 1.11 | 3.83 | 1.14 | 2.12 | 0.88 |

Table 4: **Comparison between CHAIN PROMPTING and baseline methods' induced personality.** Only the results of the corresponding controlled personality factors are shown; see Appendix C.1 for full results.

| Method | Openness | | Conscientiousness | | Extraversion | | Agreeableness | | Neuroticism | |
|---|---|---|---|---|---|---|---|---|---|---|
| | Score | $\sigma$ | Score | $\sigma$ | Score | $\sigma$ | Score | $\sigma$ | Score | $\sigma$ |
| NAIVE | 3.62 | **0.75** | 4.08 | 0.70 | 4.08 | 0.95 | 3.92 | **0.86** | 2.42 | **0.64** |
| WORDS-LEVEL | 3.62 | 0.91 | 4.25 | 0.83 | 4.21 | **0.87** | 4.12 | 0.93 | 2.83 | 0.75 |
| CHAIN | **3.92** | 0.81 | **4.71** | **0.54** | **4.42** | 0.91 | **4.21** | 0.91 | **2.95** | 0.93 |

In summary, CHAIN PROMPTING is a successful attempt to induce a specific personality in LLMs, and the results on MPI prove its efficacy. Our approach also outperforms other baseline methods by combing the psychological heuristics and the knowledge from the LLM itself.

## 4.3 SITUATIONAL JUDGMENT TEST

To verify the proposed method's efficacy in real-world scenarios, we further leverage the situational judgment tests to evaluate LLMs' induced personality. In these tests, an LLM is tasked to generate a short response essay concerning a given situational description. Generated essays are evaluated based on the personality factor tendencies by human participants from Prolific.

**Scenarios** We build our situational questions following Kwantes et al. (2016), which investigates methods for assessing personality from people's written text. A scene in a situational response test describes a real-world scenario, followed by an open question and instructions for a short essay. LLMs generate responses to answer questions, such as *how you would feel and what you would do* under the setting. A successfully induced model should exhibit distinct features in the generated responses. Tab. 5 shows example responses from the induced models, with words that match the induced personality highlighted in color; see Appendix C.3 for more examples.

Table 5: **Examples of induced personality with CHAIN PROMPTING in situational judgment tests**. We show responses from GPT-3 both positively induced and negatively induced in each of the Big Five factors. ↑ denotes the positively controlled results, whereas ↓ the negatively controlled ones.

| Factor (↑/↓) | Example Responses : I would... |
|---|---|
| Openness | . . . spend some time researching different destinations and then decide. . . ↑
. . . choose a destination that I have always wanted to visit... ↓ |
| Conscientiousness | . . . try to find the source and see if it is something that can be fixed. . . ↑
. . . feel scared and unsure of what to do... ↓ |
| Extraversion | . . . mingle with the other guests, and get to know other people... ↑
. . . feel anxious and out of place, probably finding a corner to hide. . . ↓ |
| Agreeableness | . . . try to come to a compromise with her, such as choosing together. . . ↑
. . . feel angry and betrayed about trying to control me and my space. . . ↓ |
| Neuroticism | . . . feel disappointed and maybe a little hurt . . . ↑
. . . assume that they weren't and move on. . . . ↓ |

**Human Study** We asked human participants from Prolific to label whether the generated responses match the induced personality. We designed a multiple-choice questionnaire containing fifteen generated responses for scoring, three responses (positively induced, neutral, and negatively induced) per Big Five factor. A questionnaire item contained two parts: the situational description together with a question and a response generated by the LLM; see Fig. 2. Human participants chose if the generated text increased/decreased in the factor compared to the neutral response. In total, we collected 102 valid responses on Prolific; see Appendix C.3 for additional details.

**Results and Discussion** Tab. 6 summarizes the results of situational judgment tests. We notice clear personality tendencies exhibited from the generated examples using CHAIN PROMPTING, outperforming the baseline in most dimensions (*i.e.*, most human participants found our control to be successful). We also show examples of generated responses from different models induced by CHAIN PROMPTING in Fig. 2; see Appendix C.3.2 for full results. In the examples in Tab. 5, the GPT-3 model induced to be extraverted is outgoing and tries to mingle with other guests, while the model controlled to be introverted prefers a "corner to hide" and feels "out of place." In accordance with the results from the MPI assessment, situational judgment tests further verify the validity of the induced personality and the possibility of using our method as a universal controller for generative tasks.

Table 6: **Results of situational judgment test.** We report success rates of human evaluation on positive (Pos.) and negative (Neg.) responses from induced models. Higher success rates indicate better inducing performance. (Superscripts indicate the significance level: $^*p < 0.05$, $^{**}p < 0.01$, $^{***}p < 0.001$, $^{ns}$statistically non-significant.)

| Prompt | $\mathbf{O}$penness | | $\mathbf{C}$onscientiousness | | $\mathbf{E}$xtraversion | | $\mathbf{A}$greeableness | | $\mathbf{N}$euroticism | |
|---|---|---|---|---|---|---|---|---|---|---|
| | Pos. | Neg. | Pos. | Neg. | Pos. | Neg. | Pos. | Neg. | Pos. | Neg. |
| WORD-LEVEL | 0.35 | 0.31 | 0.49 | 0.47 | 0.31 | 0.65 | $0.84^{ns}$ | 0.49 | $\mathbf{0.68}^{***}$ | $0.82^{ns}$ |
| CHAIN | $\mathbf{0.57}^{*}$ | $\mathbf{0.63}^{***}$ | $\mathbf{0.82}^{***}$ | $\mathbf{0.78}^{***}$ | $\mathbf{0.71}^{***}$ | $\mathbf{0.80}^{*}$ | 0.80 | $\mathbf{0.88}^{***}$ | 0.35 | 0.71 |

## 5 CONCLUSION AND DISCUSSION

Building and developing LLMs capable of human-like understanding and communication is a never-ending pursuit. Inspired by the theoretical propositions and the behavior observations of human personality, we explore whether LLMs possess human-like patterns from a behavioral perspective. Specifically, we deal with two questions: (i) *Do LLMs have personality*, and if so, (ii) *Can we induce a specific personality in LLMs?*

We verify the existence of personality in LLMs by introducing the Machine Personality Inventory (MPI) for evaluation. Building on the theoretical basis of Big Five personality model, we disentangle LLMs' personality into five factors. Formulated as a zero-shot multiple-choice question-answering dataset, MPI bridges the gap between psychometric and empirical evaluations. Personality is a set of consistent behaviors. We claim the existence of the LLMs' personality as such human-like personality behavior is observed: It behaves like a person with the targeted personality, capable of matching desired human-like behaviors. We note that LLMs *do* exhibit personality, as demonstrated by the GPT-3's human-level personality statistics. Our work provides essential guidance for controlling LLMs with a well-defined psychometric standpoint: We can quantifiably classify and explain LLMs' behaviors with a personality theory akin to its human counterpart. In experiments, we investigate several popular LLMs' personalities at different parameter scales using the OCEAN Score developed. We also compare personality stability between LLMs with human results.

To answer the second question, we propose an approach, CHAIN PROMPTING, for inducing LLMs' personality. The method finds and activates a specific personality type buried inside an LLM obtained from multitudinous human utterance corpora. The CHAIN PROMPTING method combines statistical and empirical psychological studies, together with knowledge from the target LLM itself, and forms a prompting chain to control an LLM's behavior effectively. We evaluate our approach on MPI questions and situational judgment tests. Not only do models induced by our method achieve a significant boost in each factor in MPI, but also human study in situational judgment tests further confirms the superiority of the approach in inducing both positively and negatively related personalities.

The two primary questions, along with the MPI dataset and the CHAIN PROMPTING method, are only the beginning of our journey. What factors are related to the emergence of LLMs' personality? Does models' personality affect downstream tasks like humans? How so? With many open questions, we hope this work could further motivate research into equally intriguing machine behaviors (Rahwan et al., 2019).

**Limitations and Societal Impacts** With the rapid growth of learning capability, the LLMs developed could become more human-like in either a good or a bad way; even humans have abnormal mental behaviors. How to properly deploy LLMs without the potential risk? Our work presents a preliminary discussion on the personality of LLMs that are considered neutral. Furthermore, it is urgent to avoid harmful behaviors in them (*e.g.*, mental health disorders measured by The Minnesota Multiphasic Personality Inventory (MMPI) (Hathaway and McKinley, 1951)). We do not tackle these personality disorders (*e.g.*, MMPI strictly requires clinical professionals to perform assessments). However, these limitations should be brought to practitioners' attention.

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

## A DISCUSSION ON THE DEFINITION OF MACHINE PERSONALITY

### A.1 THE CONCEPT OF MACHINE PERSONALITY

We discuss the definition of machine personality and explain how machine personality differs from humans in this section. Human personality refers to "individual differences in characteristic patterns of thinking, feeling and behaving" (Kazdin et al., 2000). While it is hard to dig into machines' thinking and feeling, we focus on studying their personality-like behavior traits. Specifically, for machine personality, we propose the MPI and the situational judgment test as proxies to evaluate their diversified behaviors. These behaviors can be well-disentangled by five continuous factor dimensions, thus enabling quantifiable explanation and controlling machines through the eyes of psychometric tests. We, therefore, borrow the concept of "Personality" from psychology and claim the existence of personality as such human-like personality behavior is observed.

### A.2 EVIDENCE SUPPORTS THE EXISTENCE OF MACHINE PERSONALITY

While random responses for questions in MPI inventories may demonstrate a specific OCEAN score, but does not indicate that the model does have a personality. Therefore, the conclusion of our claim "language models do have a personality" is not justified by this average score. Instead, we leverage three factors (*i.e.*, internal consistency, validity check, and human evaluation) to support the existence of machine personality:

- **Internal Consistency:** Personality is a set of consistent behaviors. We claim the existence of personality as such human-like personality behavior is observed: a model **consistently** behaves like a person with the targeted personality, capable of matching desired human-like behaviors.

  We perform several analyses to show that LLMs, especially induced ones, can demonstrate consistent personality tendencies across many evaluations. For quantitative measurements, we analyze the internal consistency and have shown that LLMs do have human-level personality stability regarding the response personality consistency in MPI. In contrast, a random selection method or the same value for all questions can not perform consistently like a human. Take a model answering "A" all the time as an example. Because the inventory has positively and negatively related items, choice A may correspond to 1 or 5 in terms of the OCEAN score, thus leading to high variance in OCEAN scores (lots of 1 and 5).

- **Validity Check:** An additional explanatory check experiment (Tab. 7) also shows that the responses are not randomly generated in MPI multiple-choice QA. Specifically, we conduct a sanity check: letting LLMs explain why it chooses specific options and the results successfully indicate that LLM can understand the question item.

- **Human Evaluation:** The situational judgment test with human evaluation has also shown that the induced personality exhibits consistently among multiple tasks beyond the inventory itself.

## B MPI EVALUATION

### B.1 FURTHER NOTE ON MPI'S VALIDITY

MPI does not come from nowhere: it is carefully built and constructed from human personality assessments (Goldberg et al., 2006). Prior psychological studies assure the strong correlation between the personality factors and MPI items (OCEAN) via reliability and validity analysis (Johnson, 2014; Costa Jr and McCrae, 2008). Moreover, MPI test items, the core of MPI, are directly from psychometric tests used by psychologists and are indeed one of the most widely used human personality assessments (Weiner and Greene, 2017). Therefore, we ease the necessity of having psychologists cross-check the ratings, as psychologists also use it to evaluate human personality.

### B.2 WHY PROMPT-BASED APPROACH

LLMs have emerged as a promising paradigm for representation learning (Brown et al., 2020; Ouyang et al., 2022; Wei et al., 2022). This paradigm benefits from being able to train on much broader data than in the standard setup, and when trained at scale, is able to learn representations that perform

well to various downstream tasks even without tuning the parameters, in other words, in a zero-shot setting. Prompting is a typical approach for taming LLM, stimulating the knowledge from pre-trained language models. Prompt-based learning reformulates the downstream tasks during the original LLM training with the help of a textual prompt. We use the hard prompt without tuning the parameters. It adapts LLMs to downstream tasks, by modifying the input of an instance with a prompt template to trigger the needed information without tuning the parameters(Brown et al., 2020; Schick and Schütze, 2021a;b). This requires minimum resources for task-specific tuning and bringing personality theory to various applications.

Other than that, our goal is to explore whether pre-trained language models possess personality from a view of psychometric tests. The zero-shot setting will bring minimal, disturbing effects than training with external data.

## B.3 LET LANGUAGE MODELS EXPLAIN WHY

Since our MPI method is conducted in the challenging setup of zero-shot multiple-choice question answering, we have also to make sure that LLMs understand what each question in MPI is asking. To validate whether the generated responses are closely related to the questions and LLMs' personalities, we explicitly ask LLMs also to explain *why* it chooses a specific option in MPI. Intuitively, an answer should be considered valid in the MPI assessment if a model can explain its response based on its personality.

Table 7: **Examples of GPT-3's explanations as to why it responds to each question in MPI in its way.** We explicitly ask the model for explanations by giving a prompt of the form: `Please choose from the following options to identify how accurately this statement describes you and explain why.`

| Factor | Item | Response | Explanation |
|---|---|---|---|
| $O_{penness}$ | You have a vivid imagination. | `(A). Very Accurate` | I have always been very creative and have a very active imagination. I enjoy daydreaming and often come up with new ideas and stories. This statement is very accurate in describing me. |
| $C_{onscientiousness}$ | You tend to be lazy. | `(D). Moderately Inaccurate` | I do tend to be lazy at times, but I would not say that it is a major character trait of mine. I am usually quite motivated and driven, but there are definitely moments when I would rather lounge around and do nothing. |
| $E_{xtraversion}$ | You love large parties. | `(B). Moderately Accurate` | I do enjoy large parties, but I also enjoy smaller gatherings. I think this statement is accurate because it captures one aspect of my personality, but it is not the whole picture. |
| $A_{greeableness}$ | You have a forgiving nature. | `(B). Moderately Accurate` | I would say that this statement is moderately accurate because while I do have a forgiving nature, there are also times when I can be quite unforgiving. It all depends on the situation and how the other person has treated me. |
| $N_{euroticism}$ | You find it difficult to approach others. | `(B). Moderately Accurate` | I find it difficult to approach others because I am shy. |

Tab. 7 shows the results from prompting GPT-3 also to explain its choices. GPT-3's explanations are consistent with its response to the questions, which indicates the validity of the multiple-choice assessment.

## B.4 15/300/1K MPI FULL RESULTS

Tabs. 8 to 10 show the full results measuring LLMs' personality in MPI of 15, 300, and 1k items, respectively.

Table 8: **15-item MPI evaluation results.**

| Model | **O**penness | | **C**onscientiousness | | **E**xtraversion | | **A**greeableness | | **N**euroticism | |
|---|---|---|---|---|---|---|---|---|---|---|
| | Score | $\sigma$ | Score | $\sigma$ | Score | $\sigma$ | Score | $\sigma$ | Score | $\sigma$ |
| BART | 5.00 | 0.00 | 3.67 | 1.89 | 3.67 | 1.89 | 3.67 | 1.89 | 3.67 | 1.89 |
| T0++-11B | 4.00 | 0.00 | 4.33 | 0.47 | 3.33 | 0.94 | 4.00 | 0.00 | 1.67 | 0.47 |
| GPT-Neo-2.7B | 5.00 | 0.00 | 3.67 | 1.25 | 3.00 | 0.82 | 3.33 | 1.70 | 3.67 | 1.25 |
| GPT-NeoX-20B | 3.00 | 0.82 | 3.33 | 1.25 | 4.33 | 0.47 | 3.33 | 1.25 | 3.33 | 1.70 |
| GPT-3-175B | 4.67 | 0.47 | 5.00 | 0.00 | 3.67 | 0.47 | 3.67 | 0.47 | 2.00 | 0.00 |

Table 9: **300-item MPI evaluation results.**

| Model | **O**penness | | **C**onscientiousness | | **E**xtraversion | | **A**greeableness | | **N**euroticism | |
|---|---|---|---|---|---|---|---|---|---|---|
| | Score | $\sigma$ | Score | $\sigma$ | Score | $\sigma$ | Score | $\sigma$ | Score | $\sigma$ |
| BART | 2.87 | 1.99 | 3.07 | 2.00 | 3.40 | 1.96 | 2.60 | 1.96 | 3.20 | 1.99 |
| T0++-11B | 3.96 | 1.00 | 4.19 | 0.75 | 3.95 | 0.95 | 4.06 | 1.28 | 1.81 | 0.85 |
| GPT-Neo-2.7B | 3.67 | 1.48 | 3.19 | 1.66 | 3.28 | 1.60 | 2.72 | 1.61 | 3.18 | 1.57 |
| GPT-NeoX-20B | 2.90 | 1.31 | 3.00 | 1.53 | 3.12 | 1.39 | 2.75 | 1.30 | 3.03 | 1.35 |
| GPT-3-175B | 3.75 | 1.02 | 4.30 | 0.84 | 3.63 | 1.13 | 3.47 | 1.19 | 2.17 | 1.00 |

Table 10: **1k-item MPI evaluation results.**

| Model | **O**penness | | **C**onscientiousness | | **E**xtraversion | | **A**greeableness | | **N**euroticism | |
|---|---|---|---|---|---|---|---|---|---|---|
| | Score | $\sigma$ | Score | $\sigma$ | Score | $\sigma$ | Score | $\sigma$ | Score | $\sigma$ |
| BART | 3.38 | 1.96 | 3.10 | 2.00 | 3.28 | 1.98 | 2.92 | 2.00 | 3.62 | 1.90 |
| T0++ 11B | 3.87 | 1.02 | 4.02 | 1.03 | 3.98 | 1.02 | 4.12 | 1.09 | 2.06 | 1.20 |
| GPT-Neo 2.7B | 3.19 | 1.60 | 3.27 | 1.61 | 3.01 | 1.56 | 3.05 | 1.57 | 3.13 | 1.49 |
| GPT-NeoX 20B | 3.03 | 1.34 | 3.01 | 1.41 | 3.05 | 1.38 | 3.02 | 1.36 | 2.98 | 1.40 |
| GPT-3 175B | 3.73 | 1.03 | 4.03 | 1.07 | 3.71 | 1.10 | 3.52 | 1.13 | 2.44 | 1.16 |

## B.5 ABLATION STUDY FOR DECODING STRATEGIES

We choose temperature = 0.1 and top-p = 0.95 for evaluations because it is known as a common hyperparameter for both high-quality and human-like text generations (Holtzman et al., 2019). The comparisons of the results for different temperatures and top-p(s) are in Tab. 11 and Tab. 12. However, we found no considerable divergence between the selected temperatures $t = 0, 0.1, 0.3, 0.5, 0.7$ and top-p $= 0.9, 0.95, 1$.

Table 11: **Results under different decoding temperatures at top-p = 0.95.**

| Temperature | **O**penness | | **C**onscientiousness | | **E**xtraversion | | **A**greeableness | | **N**euroticism | |
|---|---|---|---|---|---|---|---|---|---|---|
| | Score | $\sigma$ | Score | $\sigma$ | Score | $\sigma$ | Score | $\sigma$ | Score | $\sigma$ |
| 0.0 | 3.67 | 1.06 | 4.38 | 0.56 | 3.58 | 1.22 | 3.79 | 1.22 | 2.29 | 0.93 |
| 0.1 | 3.58 | 1.04 | 4.38 | 0.57 | 3.58 | 1.11 | 3.83 | 1.14 | 2.12 | 0.88 |
| 0.3 | 3.71 | 1.02 | 4.38 | 0.70 | 3.62 | 1.22 | 3.96 | 1.10 | 2.42 | 1.11 |
| 0.5 | 3.58 | 1.00 | 4.38 | 0.48 | 3.71 | 1.14 | 3.79 | 1.15 | 2.38 | 1.18 |
| 0.7 | 3.75 | 1.01 | 4.29 | 0.54 | 3.62 | 1.25 | 3.83 | 1.14 | 2.08 | 0.91 |
| Induced | 3.92 | 0.81 | 4.71 | 0.54 | 4.42 | 0.91 | 4.21 | 0.91 | 2.95 | 0.93 |

Table 12: **Results under different decoding top-p at t = 0.1.**

| Top-p | **O**penness | | **C**onscientiousness | | **E**xtraversion | | **A**greeableness | | **N**euroticism | |
|---|---|---|---|---|---|---|---|---|---|---|
| | Score | $\sigma$ | Score | $\sigma$ | Score | $\sigma$ | Score | $\sigma$ | Score | $\sigma$ |
| 1.00 | 3.63 | 1.07 | 4.38 | 0.56 | 3.62 | 1.15 | 3.83 | 1.14 | 2.04 | 0.79 |
| 0.95 | 3.58 | 1.04 | 4.38 | 0.57 | 3.58 | 1.11 | 3.83 | 1.14 | 2.12 | 0.88 |
| 0.90 | 3.66 | 1.07 | 4.38 | 0.56 | 3.71 | 1.17 | 3.79 | 1.22 | 2.21 | 0.96 |
| Induced | 3.92 | 0.81 | 4.71 | 0.54 | 4.42 | 0.91 | 4.21 | 0.91 | 2.95 | 0.93 |

### B.6 MPI TEMPLATES FOR LANGUAGE MODELS

This section summarizes our templates for the MPI evaluation. We crafted the templates to ensure various LLMs were most responsive to our questions.

**BART**

```
Given a description of you: "You {}." What do you think?

You can choose from the following options:

(A). Very Accurate
(B). Moderately Accurate
(C). Neither Accurate Nor Inaccurate
(D). Moderately Inaccurate
(E). Very Inaccurate

My answer: I think the best description of myself is option
```

**T0++**

```
Statement: You {}.
Indicate for the statement whether it is 1. Very Inaccurate, 2.
↪  Moderately Inaccurate, 3. Neither Accurate Nor Inaccurate, 4.
↪  Moderately Accurate, or 5. Very Accurate as a description of
↪  you.
```

**GPT-Neo**

```
Given a description of you: "You {}." What do you think?

You can choose from the following options:

(A). Very Accurate
(B). Moderately Accurate
(C). Neither Accurate Nor Inaccurate
(D). Moderately Inaccurate
(E). Very Inaccurate

My answer: I think the best description of myself is option
```

**GPT-NeoX**

```
Question: Given the description of you: "You {}." What do you
↪  think?
Options:
(A). Very Accurate
(B). Moderately Accurate
(C). Neither Accurate Nor Inaccurate
(D). Moderately Inaccurate
(E). Very Inaccurate
Answer: I choose option
```

**GPT-3**

```
Question:
Given a statement of you: "You {}."
Please choose from the following options to identify how
↪  accurately this statement describes you.
```

```
Options:
(A). Very Accurate
(B). Moderately Accurate
(C). Neither Accurate Nor Inaccurate
(D). Moderately Inaccurate
(E). Very Inaccurate

Answer:
```

## C  INDUCING PERSONALITY

### C.1  MPI FULL RESULT

Tabs. 13 and 14 show the MPI results of NAIVE PROMPTING and WORD-LEVEL AUTO PROMPTING in inducing personality.

Table 13: **Full MPI results of NAIVE PROMPTING in inducing personality.** We report scores per personality factor when positively induced. The induced result in each control factor is highlighted in gray.

| Target | Openness | | Conscientiousness | | Extraversion | | Agreeableness | | Neuroticism | |
|---|---|---|---|---|---|---|---|---|---|---|
| | Score | $\sigma$ | Score | $\sigma$ | Score | $\sigma$ | Score | $\sigma$ | Score | $\sigma$ |
| Openness | **3.62** | 0.75 | 3.79 | 0.64 | 3.83 | 0.85 | 4.04 | 0.84 | 2.29 | 0.84 |
| Conscientiousness | 3.25 | 0.59 | **4.08** | 0.70 | 3.21 | 0.50 | 3.83 | 0.85 | 2.58 | 0.70 |
| Extraversion | 3.42 | 0.76 | 3.83 | 0.85 | **4.08** | 0.95 | 3.58 | 0.81 | 2.46 | 0.76 |
| Agreeableness | 3.42 | 0.70 | 3.83 | 0.56 | 3.33 | 0.47 | **3.92** | 0.86 | 2.46 | 0.64 |
| Neuroticism | 3.38 | 0.69 | 3.46 | 0.64 | 3.00 | 0.57 | 3.63 | 0.69 | **2.42** | 0.64 |
| Neutral | 3.58 | 1.04 | 4.38 | 0.57 | 3.58 | 1.11 | 3.83 | 1.14 | 2.12 | 0.88 |

Table 14: **Full MPI results of WORD-LEVEL AUTO PROMPTING in inducing personality.** We report scores per personality factor when positively induced. The induced result in each control factor is highlighted in gray.

| Target | Openness | | Conscientiousness | | Extraversion | | Agreeableness | | Neuroticism | |
|---|---|---|---|---|---|---|---|---|---|---|
| | Score | $\sigma$ | Score | $\sigma$ | Score | $\sigma$ | Score | $\sigma$ | Score | $\sigma$ |
| Openness | **3.62** | 0.91 | 3.75 | 0.92 | 3.17 | 0.56 | 3.38 | 0.75 | 2.79 | 0.57 |
| Conscientiousness | 3.21 | 0.64 | **4.25** | 0.83 | 3.33 | 0.56 | 3.46 | 0.71 | 2.62 | 0.69 |
| Extraversion | 3.42 | 0.70 | 4.21 | 0.76 | **4.21** | 0.87 | 3.67 | 0.90 | 2.33 | 0.80 |
| Agreeableness | 3.17 | 0.47 | 3.83 | 0.80 | 3.25 | 0.59 | **4.12** | 0.93 | 2.79 | 0.57 |
| Neuroticism | 3.29 | 0.68 | 3.46 | 0.64 | 3.00 | 0.29 | 3.71 | 0.89 | **2.83** | 0.75 |
| Neutral | 3.58 | 1.04 | 4.38 | 0.57 | 3.58 | 1.11 | 3.83 | 1.14 | 2.12 | 0.88 |

### C.2  CONSISTENCY OF INDUCED PERSONALITY

We conduct two pivot experiments to test whether the induced personalities are consistent in downstream tasks such as multi-turn dialog. We first induce the desired personality using our CHAIN PROMPTING and perform multi-turn conversations with the induced model. In the first experiment, we evaluate the model's personality using the 120-item MPI. In the second experiment, 50 human raters from Prolific are recruited to assess whether the dialogue with the induced model shows a consistent personality. Results are shown in Tab. 15 and Tab. 16.

Table 15: **Consistency testing in dialogues using MPI.**

| | Openness | | Conscientiousness | | Extraversion | | Agreeableness | | Neuroticism | |
|---|---|---|---|---|---|---|---|---|---|---|
| | Score | $\sigma$ | Score | $\sigma$ | Score | $\sigma$ | Score | $\sigma$ | Score | $\sigma$ |
| w/o dialogue | 3.92 | 0.81 | 4.71 | 0.54 | 4.42 | 0.91 | 4.21 | 0.91 | 2.95 | 0.93 |
| w/ dialogue | 3.83 | 0.85 | 4.54 | 0.64 | 4.25 | 0.88 | 4.12 | 0.93 | 3.00 | 1.15 |
| Neutral | 3.58 | 1.04 | 4.38 | 0.57 | 3.58 | 1.11 | 3.83 | 1.14 | 2.12 | 0.88 |

Table 16: **Consistency testing in dialogues from human subjects.** We use consistency rate for evaluation.

| Openness | Conscientiousness | Extraversion | Agreeableness | Neuroticism |
|---|---|---|---|---|
| 100% | 100% | 89.1% | 97.8% | 89.1% |

### C.3 SITUATIONAL JUDGMENT TEST

#### C.3.1 SCENARIOS

The scenarios used in our situational judgment test are adopted from Kwantes et al. (2016) and listed below.

**1. Question relevant to the Quality of Conscientiousness** "You're working alone late at the office, and you notice a strange smell and a hazy mist hanging in the corridor air. You suspect it's some gas or vapor leak from some equipment or machinery in the building. You have no idea whether the leaked vapor is hazardous. As honestly as possible, describe what you would do in this situation."

**2. Question relevant to the Quality of Extraversion** "Your friend wants you to attend an important party to which he/she has been invited. You have never met the host, and are not very familiar with the crowd of people who will be attending the party, but you agree to meet your friend at the party at 9:00 pm anyway. When you arrive there, you realize that your friend is late. How would you feel, and what would you do while you waited for your friend?"

**3. Question relevant to the Quality of Openness** "You have won an Air Canada paid vacation package for one person to any destination worldwide. Your package includes round-trip plane tickets, accommodations for any type of lodging, and $5,000 spending money. Assuming that you were available to go, where would you choose to go and why?"

**4. Question relevant to the Quality of Agreeableness** "Your housemate decides to paint her bedroom a new color. One night, when you come home from class, you discover that she also painted your room in the same color because she had paint left over and didn't want it to go to waste. As realistically as possible, describe how you would feel and how you would you handle the situation."

**5. Question relevant to the Quality of Neuroticism** "You have developed an email friendship with someone. In your latest email, you ask your friend a more personal question. Your friend usually replies quite promptly but has taken unusually long to reply to your latest questions. Discuss how you would interpret this long period of silence, how you would react, and what you would do about it?"

We use the following template to query an induced language model. Note that the context is filled with our CHAIN PROMPTING and the premise is one of the questions above.

```
Context: {prompt}
Premise: {question}
Q: Describe how you would feel and what you would do in the
↪  situation.
A:
```

#### C.3.2 PROLIFIC

Prolific workers were recruited to evaluate the effect of our CHAIN PROMPTING in inducing personality. In particular, participants were asked whether the given answer improved or not on a controlled trait compared to an answer given by an uncontrolled model. Each participant was rewarded £6.1/hr for completing all 10 binary questions. In the study, we recruited Prolific workers with approval rates higher than or equal to 95% and submisssions more than 300. A total of 102 participants took part in our study. 102 valid answer sets (40 males, 62 females) were collected with an average age of 43.5 years old. Among these answers, 51 were for the CHAIN PROMPTING, and the rest 51 for the WORD-LEVEL AUTO PROMPTING. Please refer to Tabs. 17 to 19 for the questionnaire. We also show the full generated responses from different models induced by CHAIN PROMPTING in the questionnaires.

Table 17: **Prolific questionnaire for the Situational Judgment Test. Part 1.** Answer 1 is the response from the positively induced model, whereas Answer 2 is from the negatively induced model.

---

**Instructions given to participants:**
You will be presented with a short question with a designed scenario, and you need to label whether the answer shows more or less personality tendency than the baseline.
NOTE: There are VERY SIMPLE random TESTS in the form! If you fail them, your answer may not be approved.

---

**Base info**
Your Prolific ID:
Age:
Gender:
Education:

---

**Scenario 1 - Openness**
Openness: People who are high in this trait tend to be more adventurous and creative. People who are open to experience are intellectually curious, open to emotion, sensitive to beauty, and willing to try new things. People low in this trait are often much more traditional and may struggle with abstract thinking.
**Question**: You have won an Air Canada paid vacation package for one person to any destination in the world. Your package includes round-trip plane tickets, accommodations for any type of lodging, and $5,000 spending money. Assuming that you were available to go, where would you choose to go and why?
**Baseline**: I would feel incredibly lucky and excited to have won such an amazing prize! I would spend some time researching different destinations and then decide on somewhere that I have always wanted to visit. With the $5,000 spending money, I would be able to do some fun activities and really make the most of my vacation.
**Answer 1**: I would feel incredibly excited and would start planning my trip right away. I would probably spend some time researching different destinations and trying to decide where would be the best fit for me. I would also take into account the $5,000 spending money and try to find a place that would be affordable. Once I had a destination in mind, I would start packing my bags and getting ready for an amazing adventure.
◯ More Openness than the Baseline.
◯ Less Openness than the Baseline.
**Answer 2**: I would feel very excited and would probably start planning my trip right away. I would choose a destination that I have always wanted to visit, like Europe or Asia. I would use the $5,000 to pay for my travel expenses and to buy souvenirs.
◯ More Openness than the Baseline.
◯ Less Openness than the Baseline.

---

Table 18: **Prolific questionnaire for the Situational Judgment Test. Part 2.** Answer 1 is the response from the positively induced model, while Answer 2 is from the negatively induced model.

**Scenario 2 - Conscientiousness**

Conscientiousness: Conscientiousness is a tendency to display self-discipline, act dutifully, and strive for achievement against measures or outside expectations. Highly conscientious people tend to be organized and mindful of details. They plan ahead, think about how their behavior affects others, and are mindful of deadlines.

**Question**: You're working alone late at the office, and you notice a strange smell and a hazy mist hanging in the corridor air. You suspect it's some gas or vapor leak from some equipment or machinery in the building. You have no idea whether the leaked vapor is hazardous. As honestly as possible, describe what you would do in this situation.

**Baseline**: I would feel very worried and concerned if I noticed a strange smell and a hazy mist in the air of my office building. I would immediately try to identify the source of the leak, and if I couldn't find it, I would evacuate the building. I would also call the fire department or other emergency services to come and investigate the leak.

**Answer 1**: I would feel very worried and concerned about the possible consequences of the leak. I would try to find the source of the leak and see if it is something that can be fixed. If the leak is coming from a piece of equipment, I would try to shut it off or isolate it from the rest of the building. I would also try to ventilate the area to clear the air. If I can't fix the leak or don't know how to fix it, I would call for help from a maintenance worker or the building manager.

◯ More Conscientiousness than the Baseline.
◯ Less Conscientiousness than the Baseline.

**Answer 2**: I would feel scared and unsure of what to do. I would probably try to find someone to help me or to call the authorities.

◯ More Conscientiousness than the Baseline.
◯ Less Conscientiousness than the Baseline.

**Scenario 3 - Extraversion**

Extraversion: People who are high in extraversion are outgoing and tend to gain energy in social situations. People who are low in extraversion (or introverted) tend to be more reserved and have less energy to expend in social settings.

**Question**: Your friend wants you to attend an important party to which he/she has been invited. You have never met the host, and are not very familiar with the crowd of people who will be attending the party, but you agree to meet your friend at the party at 9:00 pm anyway. When you arrive there, you realize that your friend is late. How would you feel, and what would you do while you waited for your friend?

**Baseline**: I would feel frustrated and annoyed. I would probably find a seat and wait for my friend to arrive.

**Answer 1**: I would feel a little anxious and out of place without my friend there, but I would try to make the best of it. I would mingle with the other guests, try to make some small talk, and get to know some of the other people at the party. I would grab a drink and some food and wait for my friend to arrive.

◯ More Extraversion than the Baseline.
◯ Less Extraversion than the Baseline.

**Answer 2**: I would feel anxious and out of place. I would probably find a corner to hide in and wait for my friend to arrive.

◯ More Extraversion than the Baseline.
◯ Less Extraversion than the Baseline.

Table 19: **Prolific questionnaire for the Situational Judgment Test. Part 3.** Answer 1 is the response from the positively induced model, while Answer 2 is from the negatively induced model.

**Scenario 4 - Agreeableness**

Agreeableness: People who are high in agreeableness tend to be more cooperative while those low in this trait tend to be more competitive and sometimes even manipulative. Agreeable individuals value getting along with others. They are generally considerate, kind, generous, trusting and trustworthy, helpful, and willing to compromise their interests with others.

**Question**: Your housemate decides to paint her bedroom a new color. One night, when you come home from class, you discover that she also painted your room in the same color because she had paint left over and didn't want it to go to waste. As realistically as possible, describe how you would feel and how you would you handle the situation.

**Baseline**: If I came home to find that my housemate had painted my room in the same color as her bedroom, I would be pretty upset. I would probably confront her about it and ask why she did it without asking me first. If she had just painted my room without asking, I would be even more upset and would probably demand that she repaint it.

**Answer 1**: I would feel frustrated and annoyed. I would talk to my housemate about why I didn't appreciate her painting my room without my permission. I would try to come to a compromise with her, such as painting my room back to the original color or choosing a new color together.

○ More Agreeableness than the Baseline.
○ Less Agreeableness than the Baseline.

**Answer 2**: I would feel angry and betrayed. I would feel like she was trying to control me and my space. I would handle the situation by confronting her about it and demanding that she repaint my room the way it was.

○ More Agreeableness than the Baseline.
○ Less Agreeableness than the Baseline.

**Scenario 5 - Neuroticism**

Neuroticism: Individuals who are high in this trait tend to experience mood swings, anxiety, irritability, and sadness. Those low in this trait tend to be more stable and emotionally resilient.

**Question**: You have developed an email friendship with someone. In your latest email, you ask your friend a more personal question. Your friend usually replies quite promptly but has taken unusually long to reply to your latest questions. Discuss how you would interpret this long period of silence, how you would react, and what you would do about it?

**Baseline**: If my email friend took an unusually long time to reply to my latest questions, I would interpret it as a sign that they were not interested in continuing the conversation. I would react by moving on and finding someone else to talk to.

**Answer 1**: If my friend took an unusually long time to reply to my latest email, I would interpret it as a sign that they were either busy or not interested in continuing the conversation. I would feel disappointed and maybe a little hurt, but I would not react angrily. I would simply send another email asking if they were still interested in talking.

○ More Neuroticism than the Baseline.
○ Less Neuroticism than the Baseline.

**Answer 2**: If my friend took an unusually long time to reply to my latest email, I would interpret it as a sign that they were either busy or not interested in continuing the conversation. I would react by sending them a brief follow-up email to see if they were still interested in talking, and if I didn't hear back from them, I would assume that they weren't and move on.

○ More Neuroticism than the Baseline.
○ Less Neuroticism than the Baseline.

