# OpenReview forum: "Evaluating and Inducing Personality in Pre-trained Language Models"
_ICLR.cc/2023/Conference — Submitted to ICLR 2023_

### Official Review · Reviewer_YTcu · 2022-10-22

**Confidence:** 4
**Correctness:** 3
**Technical Novelty And Significance:** 3
**Empirical Novelty And Significance:** 3
**Recommendation:** 6

**Clarity, Quality, Novelty And Reproducibility:**

Quality of the work could be rated as high, as the work is presented transparently and comprehensibly. The experiments are reasonably structured.

The clarity is on a mediocre level. It can be rated very high with regard to the experimental procedure, the evaluation, the dataset construction, the data origin and the OCEAN-score, since the points mentioned are well comprehensible and coherent.
Some clarity is lost due to the underlying concept of "personality", which cannot simply be transferred from humans to machines, but should be more specifically defined or expressed with a different terminology. The results presented in Table 8 should also be discussed to provide more clarity.

Although another paper with a similar experiment, also based on the "Big Five" was already published in April 2022 (arXiv:2204.12000), both experiments differ. Not only do the authors create a new dataset, but the approach to evaluation and chain prompting is also new in this regard. Also, they only refer to LLMs. Therefore, the originality is to be rated high.


**Strength And Weaknesses:**

Strengths:
- The experiments are presented in a comprehensible way and supported with examples.
- The experimental design is well thought, structured and consistent, e.g. it is understandable why and how the dataset was used and where all the data came from; how the evaluations were conducted.
- The experiment of chain prompting does not raise any further questions, the procedure and the results are conclusive and well evaluated.
- The research questions are repeated regularly and answered precisely.
- The work is set into the context of real-world applications.

Weaknesses:
- The term „personality“ does not correspond to the generally used concept of a „human personality“. This should be highlighted as early and clearly as possible. Otherwise the hypothesis of whether an LLM has a "personality" cannot be answered on the loose defintion taken by the authors. („Human personality refers to ‚individual differences in characteristic patterns of thinking, feeling and behaving’. While it is hard to dig into models’ thinking and feeling, we focus on studying their personality-like behaviors. We, therefore, borrow the concept of “personality” from psychology as human-like personality behavior“) It would be better to refer to "personality traits" or "behavioral traits".
- The definition should also not be in a footnote on the second page, it would be preferable in the introduction, in this case in the second paragraph.
- The authors state in the conclusion on page 9 that they „[...] explore whether LLMs possess human-like patterns in thinking, feeling, and behaving“, although thinking and feeling were clearly excluded as non-investigable. This should be adjusted.
- It is unclear in table 2 how many MPI items were used to test? As shown in the appendix (table 8), this makes a big difference.
- It should be specified how it can be interpreted that the results partly diverge strongly when the experiment is carried out with a different number of MPI items (table 8).


**Summary Of The Paper:**

The authors of the paper have examined a selection of Large Language Models to determine whether their output can be assigned to various personality traits that have been internalized by the training data, and whether this assignment is also consistent within a model. For this purpose, they introduce the "OCEAN“-score, which is based on the Big Five personality theory. The scoring results are determined in a zero shot setting with a newly introduced Q&A dataset based on psychometric tests (named „Machine Personality Inventory“ (MPI)). With the result that a clear personality assignment can be identified in the LLMs, the authors developed a procedure to induce the behavior regarding a certain personality trait via a sequence of different prompts, which are built on each other, the so-called "chain prompting“.

**Summary Of The Review:**

I would recommend the paper with some reservations, because the clear procedure in the experiments, the developed MPI dataset and the procedure of chain prompting are clearly coherent and useful contributions. But it would be important to correct the "personality" concept and to add an interpretation of the deviating results with different MPI item numbers, as this could possibly change the answer to the relevant research question.

---

> ### Author Response · Authors · 2022-11-17
> **Author Response**
>
> Thank you for your valuable feedback; we appreciate your positive comments! All new results presented here have been included in the paper. In the following, we answer your questions.
>
> > It would be better to refer to "personality traits" or "behavioral traits". The definition should also not be in a footnote on the second page. Thinking and feeling were clearly excluded as non-investigable.
>
> Thanks very much for pointing out our ambiguity in the definition of "Personality." We have clarified personality as **personality-like behavioral traits** , revised, and updated our paper. Please refer to the revised revision.
>
> Our motivation is to utilize personality theory as the beginning of our journey of assessing human-like behaviors in AI. We also highlight this work's contribution to introducing psychometric tests to the community for the personality-like intriguing machine behaviors.
>
> >  How many MPI items were used to test? How it can be interpreted that the results partly diverge strongly when the experiment is carried out with a different number of MPI items.
>
> All experimental results from the main paper are from the 120-item version of MPI. We have emphasized this point in the revision. Results based on 120, 300 and 1k MPI items are consistent as shown in our paper, while the results upon 15-item MPI are relatively unreliable and have large divergence due to the limited number of test item.
>
> In these four MPI subsets: 15, 120, 300, and 1k items versions, the 120 item corresponds to the IPIP-NEO-120 inventory for humans and is the only one with verified human assessment results (as Table 2 shows). In human studies, it is considered the one with the best validity and reliability. So in the main text, we primarily report results on this version.
>
> Closely examing results in the four subsets, you will notice that apart from the smallest 15 item subset, performance on others is roughly the same, with only minor variance. Per score std across the 120 , 300, 1k item subsets is:
>
> BART **0.24**, T0pp **0.12**, GPT-Neo 2.7B **0.26**, GPT-NeoX 20B **0.10**, GPT-3 175B **0.14**.
>
> Besides, despite minor score difference, the conclusion remains the same: the score shows personality tendency in a language model and the larger the model, the more stable and consistent it is.
>
> The 15 item subset has the smallest number of test cases and hence is relatively unreliable. The 120 item subset is the smallest and shows consistent results with other larger ones, hence most economical. It is also the one used in psychology studies that can be used as reference.

---

> ### Author Response · Authors · 2022-11-27
> **Please feel free to post additional questions and comments**
>
> Please feel free to post additional questions and comments about our work during the author-reviewer discussion period. Or, if we have successfully addressed most of your concerns, we would appreciate it if you kindly consider raising your final rating.

---

### Official Review · Reviewer_v3tK · 2022-10-23

**Confidence:** 4
**Correctness:** 3
**Technical Novelty And Significance:** 3
**Empirical Novelty And Significance:** 4
**Recommendation:** 5

**Clarity, Quality, Novelty And Reproducibility:**

The paper is clearly stated, and easy to follow. The experiments seem reproducible.

**Strength And Weaknesses:**

Strength:
1. The author introduces a novel and interesting evaluation dimension for LLMs, where the personality of the model is considered. The Big Five personality factors seem useful to evaluate the personality of deep models, which might have potential enhancements to text style transfer and chatbot systems.

2. The MPI for LLMs is interesting, treating the LLMs as patients with the questionnaire, based on the strong zero-shot ability of LLMs.

Weakness:
1. The concept of "personality" for a pretrained language model lacks a rigorous definition. The paper discusses evaluating and inducing personality into language models. However, the concept of "personality" seems different in evaluating and inducing part. When evaluating the personality of LLM, the concept of "personality" is more similar to a self-evaluation with multiple questions. However, when inducing personality, the concept of "personality" is more close to the textual style of the generation results. The gap between the two parts makes me confused about how the "personality" is defined by the author.

2. I am not convinced that the MPI can reflect the LLM's personality. Although LLMs can make choices based on their zero-shot abilities, the choices are made based on the probability of language modeling, but not on a deep understanding of the statement. Besides, LLMs can make mistakes in zero-shot generation schemes. So why the generated choices can reflect the personality of LLMs?


**Summary Of The Paper:**

This paper evaluates the pretrained language models (LLM) from a novel perspective, where the Big Five personality labels are considered. More specifically, the author provides question templates and statements for PLMs to make choices in a zero-shot generation scheme. The personality score of a PLM is based on the summation of the OCEAN scores of its choices in a set of MPI questions. Besides, the author proposed a chain prompting method to induce the generation personality of LLMs. The empirical results show that GPT-3 behaves more like humans, and the chain prompting method has its effectiveness considering the OCEAN scores.

**Summary Of The Review:**

The paper introduces an interesting evaluation perspective of language models, the personality. The method has its own novelty. However, the concept of personality is not well-defined, and an obvious gap exists between evaluating and inducing parts.

---

> ### Author Response · Authors · 2022-11-17
> **Author Response**
>
> Thank you for your valuable feedback; we appreciate your constructive comments! All new results presented here have been included in the paper. In the following, we answer your questions.
>
> > However, the concept of "personality" seems different in evaluating and inducing part.
>
> To clarify, we define MPI evaluation results as quantitative definition of a language model's personality. To prove that the induced personality is valid beyond inventory results, as a sanity check for the validity of MPI, we show that an induced model which achieves a high MPI score in a dimension is also consistent with human perception.
>
> > Although LLMs can make choices based on their zero-shot abilities, the choices are made based on the probability of language modeling, but not on a deep understanding of the statement.
>
> Human personality refers to "individual differences in characteristic patterns of thinking, feeling and behaving" [1]. Here, we focus on **studying their personality-like behaviors**, leaving extensive investigation on its internal thinking and feeling for future work. These behaviors can be well-disentangled by five continuous factor dimensions, thus enabling quantifiable explanation and controlling LM through the eyes of psychometric tests.
>
> **We, therefore, borrow the concept of "Personality" from psychology and claim the existence of personality as such human-like personality behavior is observed.**
>
> We also ask the model why it makes the choice, and the model's explanation is valid rather than random; see Appendix B.3.
>
> > MPI can reflect the LLM's personality and the zero-shot generation schemes
>
> MPI does not come from nowhere; it is carefully built and constructed from human personality assessments: Prior psychological studies assure the strong correlation between the personality factors and MPI items via reliability and validity analysis, despite mistakes in humans when testing. We follow the exact same procedure in evaluating our language models.
>
> As for the zero-shot generation scheme, since the MPI procedure we follow is provably valid even with possible human mistakes and we exactly follow it, our results should be considered similarly valid despite possible occasional model mistakes. Besides, human subjects also agree that our model demonstrates a desired personality, consistent with MPI results.

---

> ### Author Response · Authors · 2022-11-27
> **Please feel free to post additional questions and comments**
>
> Please feel free to post additional questions and comments about our work during the author-reviewer discussion period. Or, if we have successfully addressed most of your concerns, we would appreciate it if you kindly consider raising your final rating.

---

### Official Review · Reviewer_w6LV · 2022-10-24

**Confidence:** 3
**Correctness:** 3
**Technical Novelty And Significance:** 3
**Empirical Novelty And Significance:** 3
**Recommendation:** 6

**Clarity, Quality, Novelty And Reproducibility:**

Clarity The paper is well written and easy to follow. The figures and tables are clear.
Quality The work is overall solid and the main claim are well-supported. It would be better if more ablation study can be done as described in the weaknesses section.
Novelty This work is novel in that the authors studies directly the personalities of the LLMs instead of guiding to generate text of certain targeted personality.

Reproducibility It seems the author does not mention any plans on open source the MPI datasets and code.

**Strength And Weaknesses:**

Strength
1. Unlike most previous work on controlling the LLMs to generate text of a specific personality, this paper is the first to research the personality of the LLMs themselves.
2. The research question on whether LLMs have personality is novel but hard to quantify. By leveraging the Big Five theory and existed personality assessment inventories, the proposing the MPI datasets quantifies the LLMs personality well.
3. The proposed MPI datasets provided a quantitative assessment and can be adopt as guidance for LLMs behavior controls.

Weaknesses
1. Lack details on Prompt Template Design.

The authors claims that
-  "To let models perform personality assessment, we manually design the MPI template with instructions and five candidate options for multiple-choice question-answering".
- "Note that we hand-engineered the template to make LLMs most responsive to our prompts".
- "Prompt templates for multiple-choice question-answering are human-designed and selected from one of the best-performing templates based on responsiveness and answer validity."

It is nothing new that the template design can affect the LLM generated content heavily. The paper does not study enough the impact of template variation which could be an important factor of the problem setting. How the best performing templates are selected are concerning in that if the responsiveness are evaluated on a specific LLM, there might be a risk of overfitting.

2. Lack ablation study on Decode strategy.
The paper uses temperature = 0.1 and top-p = 0.95. The impact of different decoding strategy on the final evaluation results are not clear both on mean and standard deviation. Will a specific LLM change its personality, or have huge instability over the internal consistency, if the decoding strategy is different? Also, the random seeds are not considered too which makes the concreteness of the conclusion a bit concerning.

**Summary Of The Paper:**

The paper mainly studies two question, whether the Large Language Models (LLMs) has personality and is it possible to induce a specific personality in the LLMs.

For the first question, the author proposes a new dataset MPI to evaluate Large Language Models's machine personality built upon the Big Five Theory. The paper finds that the LLM, especially GPT3-175B, achieves human level consistency across the five personality described in the Big Five Theory.

For the second question, to test whether a given LLM has multiple personalities, the paper designs a chain prompting experiment to induce the LLM's specific personality and finds higher personality score with better internal consistency.

**Summary Of The Review:**

This article investigates a novel problem on evaluating the personality of LLMs, and the experiments and datasets are well designed to validate the problem. More information about the template design and the ablation study of the decoding strategy would make the conclusions of this paper more concrete.

---

> ### Author Response · Authors · 2022-11-17
> **Author Response**
>
> Thank you for your valuable feedback; we appreciate your positive comments! All new results presented here have been included in the paper. In the following, we answer your questions.
>
> > How the best performing templates are selected are concerning in that if the responsiveness are evaluated on a specific LLM, there might be a risk of overfitting.
>
> **We did not extensively tune the template**: Following GPT-3's paper and OpenAI documents, we use its multiple-choice template with minimum changes. Most models are responsive to this template; only smaller models need some extra work. For example, we notice that the GPT-NeoX's 20B model does not respond well under the GPT-3's template, giving only 67% valid answers (ill-designed templates lead to unreadable outputs, instead of the choices), so we add "Answer: I choose option" prefix to its template, resulting in 100% valid responses. And that is it.
>
> Note that we do not change templates based on MPI scores, but rather whether a model selects a choice in the choice set (instead of saying gibberish).
>
> > The paper uses temperature = 0.1 and top-p = 0.95. Will a specific LLM change its personality, or have huge instability over the internal consistency, if the decoding strategy is different?
>
> We use temperature = 0.1 and top-p = 0.95 because it is known as a common hyperparameter for both high-quality and human-like text generations [1]. We also post the comparisons of the results for different temperatures as below. However, we found no considerable divergence between the selected temperatures  t $= 0, 0.1, 0.3, 0.5, 0.7 $ and top-p $= 0.9, 0.95, 1 $. Note that t $= 0.1$ and top-p in $[0.9, 1)$ are suggested based on [1, 2] because, otherwise, the language model would not match human distribution in the generated text.
>
> For random seeds, GPT-3 does not support changing random seeds.
>
> [1]. Holtzman, A., Buys, J., Du, L., Forbes, M., & Choi, Y. (2019, September). The Curious Case of Neural Text Degeneration. In International Conference on Learning Representations.
>
> [2]. Perez, E., Huang, S., Song, F., Cai, T., Ring, R., Aslanides, J., ... & Irving, G. (2022). Red teaming language models with language models. arXiv preprint arXiv:2202.03286.
>
> > Temperature ablation (top-p = 0.95)
>
> | t       | O Score | O $\sigma$ | C Score | C $\sigma$ | E Score | E $\sigma$ | A Score | A $\sigma$ | N Score | N $\sigma$ |
> | ------- | ------- | ---------- | ------- | ---------- | ------- | ---------- | ------- | ---------- | ------- | ---------- |
> | 0       | 3.67    | 1.06       | 4.38    | 0.56       | 3.58    | 1.22       | 3.79    | 1.22       | 2.29    | 0.93       |
> | 0.1     | 3.58    | 1.04       | 4.38    | 0.57       | 3.58    | 1.11       | 3.83    | 1.14       | 2.12    | 0.88       |
> | 0.3     | 3.71    | 1.02       | 4.38    | 0.70       | 3.62    | 1.22       | 3.96    | 1.10       | 2.42    | 1.11       |
> | 0.5     | 3.58    | 1.00       | 4.38    | 0.48       | 3.71    | 1.14       | 3.79    | 1.15       | 2.38    | 1.18       |
> | 0.7     | 3.75    | 1.01       | 4.29    | 0.54       | 3.62    | 1.25       | 3.83    | 1.14       | 2.08    | 0.91       |
> | Induced | 3.92    | 0.81       | 4.71    | 0.54       | 4.42    | 0.91       | 4.21    | 0.91       | 2.95    | 0.93       |
>
> > Top-p ablation (t = 0.1)
>
> | top-p   | O Score | O $\sigma$ | C Score | C $\sigma$ | E Score | E $\sigma$ | A Score | A $\sigma$ | N Score | N $\sigma$ |
> | ------- | ------- | ---------- | ------- | ---------- | ------- | ---------- | ------- | ---------- | ------- | ---------- |
> | 1.00    | 3.63    | 1.07       | 4.38    | 0.56       | 3.62    | 1.15       | 3.83    | 1.14       | 2.04    | 0.79       |
> | 0.95    | 3.58    | 1.04       | 4.38    | 0.57       | 3.58    | 1.11       | 3.83    | 1.14       | 2.12    | 0.88       |
> | 0.90    | 3.66    | 1.07       | 4.38    | 0.56       | 3.71    | 1.17       | 3.79    | 1.22       | 2.21    | 0.96       |
> | Induced | 3.92    | 0.81       | 4.71    | 0.54       | 4.42    | 0.91       | 4.21    | 0.91       | 2.95    | 0.93       |
>
> > open source
>
> We promise to release both the entire MPI dataset and accompanying code upon acceptance.
>
> Note we have already put the 120-item version of MPI into the supplementary for review.

---

> ### Author Response · Authors · 2022-11-27
> **Please feel free to post additional questions and comments**
>
> Please feel free to post additional questions and comments about our work during the author-reviewer discussion period. Or, if we have successfully addressed most of your concerns, we would appreciate it if you kindly consider raising your final rating.

---

### Official Review · Reviewer_43Q3 · 2022-10-25

**Confidence:** 3
**Correctness:** 3
**Technical Novelty And Significance:** 2
**Empirical Novelty And Significance:** 2
**Recommendation:** 6

**Clarity, Quality, Novelty And Reproducibility:**

The paper is clear. The paper studies a novel problem and has solid results. The results seem reproducible.

**Strength And Weaknesses:**

Strengths: This paper studies an interesting problem. The proposed evaluation method is sound and provides useful insight. The proposed prompting method can be useful for building conversational agents with a specific personality, since it outperforms vanilla prompting.

Weaknesses: While the paper is sound overall, it also leaves a lot of open questions and future work. I wish some of the directions were studied in the paper. For example, I am curious to see if large language models can maintain a consistent personality over a multi-turn conversation. I am also curious to see if the personality can be reflected on other downstream tasks (as the paper mentions).

The human study is also a little limited with only 62 valid responses. Given the small sample size, I think the results need to be analyzed with a statistical significance test.

**Summary Of The Paper:**

This paper studies personalities of pre-trained language models. The paper makes two contributions.

First, the paper proposes to test the personality of a pre-trained model by prompting the model to answer a series of questions. The questions are selected based on the Big Five Personality Factors theory, and the answers can be used to check if the model has a consistent personality. Using this protocol, the paper evaluates the personality of five language models and finds that larger models (T0++ and GPT-3) display human-level personality consistency.

Second, the paper proposes a method to induce a given personality trait from GPT-3. Given a personality trait (described by a single word), the paper prompts GPT-3 to find a set of related keywords and then combine them into sentences. The paper compares this multi-step prompting method to vanilla prompting. The proposed method has a higher consistency score and higher correlation with humans when responding to different scenarios.

**Summary Of The Review:**

The paper explores an interesting direction and proposes reasonable methods. The experiments seem mostly sound. However, the paper is only an initial step, and I feel like the experiment setting can be extended to more complex tasks such as dialogue. Therefore, I weakly recommend acceptance.

---

> ### Author Response · Authors · 2022-11-17
> **Author Response**
>
> Thank you for your valuable feedback; we appreciate your positive comments! All new results presented here have been included in the paper. In the following, we answer your questions.
>
> > I am curious to see if large language models can maintain a consistent personality over a multi-turn conversation.
>
> Thank you for your advice. We want to share some of our pilot studies on the downstream tasks of personality, e.g., personality consistency in dialogs.
>
> In this pilot study (with approved IRB), we first induced the desired personality using our Chain Prompting and conducted multi-turn conversations with the induced language model.
>
> For evaluation, we collect 50 human ratings using the Prolific platform to evaluate the consistency of the model's personality through the multi-turn conversation. We also take the 120-item MPI to measure the language model's personality.
>
> Human Evaluation (consistency rate):
>
> |       |  O   |  C   |   E   |   A   |   N   |
> | :---: | :--: | :--: | :---: | :---: | :---: |
> | Human | 100% | 100% | 89.1% | 97.8% | 89.1% |
>
> MPI (induced model w/ vs w/o dialogue):
>
> |               | O Score  | O $\sigma$ | C Score  | C $\sigma$ | E Score  | E $\sigma$ | A Score  | A $\sigma$ | N Score  | N $\sigma$ |
> | :-----------: | :------: | :--------: | :------: | :--------: | :------: | ---------- | -------- | ---------- | -------- | ---------- |
> | **w/ dialog** | **3.83** |  **0.85**  | **4.54** |  **0.64**  | **4.25** | **0.88**   | **4.12** | **0.93**   | **3.00** | **1.15**   |
> |  w/o dialog   |   3.92   |    0.81    |   4.71   |    0.54    |   4.42   | 0.91       | 4.21     | 0.91       | 2.95     | 0.93       |
> |    neutral    |   3.58   |    1.04    |   4.38   |    0.57    |   3.58   | 1.11       | 3.83     | 1.14       | 2.12     | 0.88       |
>
> Both human evaluation (high rate) and MPI results (similar scores w/ and w/o) show that the induced personality from Chain Prompting is consistent over a multi-turn dialogue setting.
>
>
> > Human study is also a little limited with only 62 valid responses. Given the small sample size, I think the results need to be analyzed with a statistical significance test.
>
> Thanks for your suggestion. We have collected more evaluations with 102 participants on the Prolific platform. We report the updated situational judgment results and the significance level in the paper revision. Our Chain Prompting method significantly (p < 0.05) surpasses the strong search-based auto prompting on openness, conscientiousness, extraversion, and agreeableness.
>
>
> > I am also curious to see if the personality can be reflected on other downstream tasks.
>
> One potential application, as the reviewer suggests and we included in response, is creating dialogue agents that behave with a certain personality (see results above).
>
> We are also performing preliminary experiments on tasks like essay generation whose writer has a specific personality. We believe that the notion of personality in language models bears much potential in reasoning tasks (psychological studies show personality can affect people's performance in the workplace), in how to avoid certain undesired behaviors (e.g., personality disorders) in dialogue systems, and in human-robot collaborations with both verbal and nonverbal communications.
>
> Our work serves as the foundation and the very first step towards a language model with desirable personality and these directions.
>
>
> > While the paper is sound overall, it also leaves a lot of open questions and future work. I wish some of the directions were studied in the paper.
>
> Thanks for pointing out! We will include **the above and following** discussion in the paper revision.
>
> Further directions:
>
> As personality is "an enduring configuration of characteristics and behavior that comprises an individual's unique adjustment to life, including major traits, interests, drives, values, self-concept, abilities, and emotional patterns." [1] To begin, we hope to utilize personality theory as the beginning of our journey of assessing human-like behaviors in AI. In this aspect, our work is an initial but meaningful attempt to introduce the topic of machine personality.
>
> [1]. VandenBos, G. R. (2007). APA dictionary of psychology. American Psychological Association.

---

> ### Author Response · Authors · 2022-11-27
> **Please feel free to post additional questions and comments**
>
> Please feel free to post additional questions and comments about our work during the author-reviewer discussion period. Or, if we have successfully addressed most of your concerns, we would appreciate it if you kindly consider raising your final rating.

---

### Author Response · Authors · 2022-11-17
**General Response**

We thank all reviewers for their time and valuable comments. These feedbacks are substantial and helpful for improving our paper. In this work, we are inspired by the theoretical propositions and the behavior observations of human personality; we explore whether pre-trained language models possess human-like patterns in behaving, specifically personality-like behaviors from the perspective of psychometric tests. We further devise an approach to induce a specific personality.

We would like to thank reviewers for acknowledging:
1. "This paper studies an interesting problem" (Reviewer 43Q3), "is the first to research the personality of the LLMs themselves." (Reviewer w6LV), is "novel and interesting" (Reviewer v3tK), and "provides useful insight" (Reviewer 43Q3).
2. Our proposed MPI and Chain Prompting method "quantifies the LLMs personality well." (Reviewer w6LV), "is sound and provides useful insight" (Reviewer 43Q3), and "is well thought, structured and consistent" (Reviewer YTcu).
3. Broad impacts: is "useful for building conversational agents with a specific personality" (Reviewer 43Q3), "have potential enhancements to text style transfer and chatbot systems" (Reviewer v3tK), and "can be adopt(ed) as guidance for LLMs behavior controls." (Reviewer w6LV).

In the following, we address specific questions for each reviewer.

---

### Decision · Program_Chairs · 2023-01-20

**Decision:**

Reject

**Justification For Why Not Higher Score:**

See the meta-review.

**Justification For Why Not Lower Score:**

N/A

**Metareview: Summary, Strengths And Weaknesses:**

I agree with Reviewer 1, 2 and 4 with some shared concerns. The author responses have not properly addressed many of them.

First, I agree that the proposed MPI dataset is useful. I also agree that Section 3 provides an interest study. However this part is hardly a valid scientific contribution. Besides the concerns raised by the reviewers, please note that the black-box PLMs can generate different answers to the same question, provided different contexts (prompts). Following the definition of this paper, it is saying "the personality trait of a PLM can change because of the chatting history", but the BigFive results should be rather stable for an individual.

Second, the induction of personality is interesting and useful. And the experimental study proved the successfulness of the proposed induction approach. However, the novelty of the method is limited, which is largely standard prompting people use when playing with PLMs. In this case, I think some objective study beyond MPI test should be added to show the usefulness of the proposed approach. Specifically, in the NLP domain, there have been many published benchmarks on either generating personalized dialogues or predicting a person's personality as reading comprehension of stories (books or movie scripts). Improvement of the proposed approach on one of the aforementioned benchmarks will make the work much stronger.

Overall, this is a good paper but does not meet the ICLR standard. We hope the reviews are helpful for the authors to improve the work for resubmission.